# Efficient polygenic risk scores for biobank scale data by exploiting phenotypes from inferred relatives

Buu Truong [1,2,3], Xuan Zhou[3,4], Jisu Shin [3,4], Jiuyong Li [1], Julius H. J. van der Werf[5], Thuc D. Le [1✉] & S. Hong Lee [3,4✉]

Polygenic risk scores are emerging as a potentially powerful tool to predict future phenotypes of target individuals, typically using unrelated individuals, thereby devaluing information from relatives. Here, for 50 traits from the UK Biobank data, we show that a design of 5,000 individuals with first-degree relatives of target individuals can achieve a prediction accuracy similar to that of around 220,000 unrelated individuals (mean prediction accuracy = 0.26 vs. 0.24, mean fold-change = 1.06 (95% CI: 0.99-1.13), P-value = 0.08), despite a 44-fold difference in sample size. For lifestyle traits, the prediction accuracy with 5,000 individuals including first-degree relatives of target individuals is significantly higher than that with 220,000 unrelated individuals (mean prediction accuracy = 0.22 vs. 0.16, mean fold-change = 1.40 (1.17-1.62), P-value = 0.025). Our findings suggest that polygenic prediction integrating family information may help to accelerate precision health and clinical intervention.

[1] UniSA STEM, University of South Australia, Mawson Lakes, SA 5095, Australia. [2] Pham Ngoc Thach University of Medicine, Ho Chi Minh City, Vietnam. [3] Australian Centre for Precision Health, University of South Australia Cancer Research Institute, University of South Australia, Adelaide, SA 5000, Australia. [4] UniSA Allied Health and Human Performance, University of South Australia, Adelaide, SA 5000, Australia. [5] School of Environmental and Rural Science, University of New England, Armidale, NSW 2351, Australia. ✉email: Thuc.Le@unisa.edu.au; hong.lee@unisa.edu.au

Genome-wide association studies (GWAS) have uncovered many common variants associated with complex traits[1,2]. In a standard GWAS, such associations are usually evaluated for many genome-wide single-nucleotide polymorphisms (SNPs), one at a time, based on data from a large number of individuals. For most complex traits and diseases, the effects of a single SNP are small, and the proportion of phenotypic variance explained by genome-wide significant SNPs is likewise small[3,4]. Therefore, an increasing interest lies in the prediction of future phenotypes for such traits from combined effects of a large number of genome-wide SNPs, as known as a whole-genome approach to genetic prediction[5–8]. There have been a number of such approaches that jointly model all or most of common variants across the genome. For instance, genome-based residual maximum likelihood (GREML)[9,10] can be used to estimate SNP-based heritability, i.e., the proportion of phenotypic variance explained by genome-wide SNPs. Best linear unbiased prediction (BLUP) can fit a genomic relationship matrix to estimate the genetic effect on the phenotype of each individual, and this method has been termed genomic BLUP (GBLUP)[11–15]. Linkage disequilibrium score regression (LDSC)[16] use aggregated effects from GWAS summary statistics of genome-wide SNPs to estimate SNP-based heritability and predict the future phenotypes of target sample for complex traits[11,17–20].

Most existing GWAS use population-based designs, in which close relatives are typically excluded or devalued from the analyses to avoid bias due to common family effects, i.e., biased SNP effects or inflated SNP-based heritability due to confounding between additive genetic and family effects. Especially when estimating narrow-sense heritability based on the genome-wide SNPs, individuals with pairwise genomic relationships >0.05 are usually excluded[4,21,22]. This convention has generally been extended to genomic prediction studies, which use similar population-based designs as GWAS[21,23,24]. However, the purpose of prediction should be clearly distinguished from that of heritability estimation. The aim of genomic prediction is to maximise phenotypic prediction accuracy. Unlike for the estimation of heritability, it is not critical to disentangle additive genetic effects from other common family effects in a genomic prediction context. In fact, such family effects could be a valuable source of information to improve prediction accuracy. Therefore, excluding close relatives for phenotypic prediction may not be well justified.

Theoretical studies have demonstrated that information from close relatives could improve prediction accuracy, even in the absence of familial environmental effects[25–28]. In these studies, it was shown that prediction accuracy depends on the effective number of chromosome segments ($M_e$), also known as the number of independent SNPs. $M_e$ is a function of effective population size ($N_e$) and it decreases when the number of high relationships between reference and target samples increases, which improves the phenotypic prediction accuracy[28]. Several studies have shown that family information is useful for polygenic risk prediction[29–31]. However, to our knowledge, there is no large-scale study to verify the efficiency of using relatives in polygenic risk prediction and its implications for clinical practice.

To predict the polygenic risk scores (PRS) of a new person for which we have DNA data, all available data in the biobank should be utilised, including any phenotypes of individuals that have a pedigree relationship with that person. Here, we use UK Biobank data and show an efficient polygenic prediction when we use relatives in a PRS approach to predict phenotypes of complex traits. We perform GWAS for 50 human complex traits including 12 disease traits using genotype and phenotype data in the reference data and use the estimated SNP effects to obtain PRS for the target sample. We investigate the contribution of information from the relatives of the target sample in polygenic risk prediction. The 50 traits are further categorised into three groups of mental, physical and lifestyle traits, and we assess the prediction performance for each group as family effects varies between the categorised traits. In addition, we extend our approach to integrate phenotypes of ungenotyped relatives of the target sample to explore whether this can further increase the prediction accuracy. We show that the efficiency of polygenic prediction with close relatives, despite a 44-fold lower in sample size, is equivalent or even higher (depending on traits) than that with unrelated individuals. This result suggests that polygenic prediction integrating family information will be a useful tool for precision health and preventive medicine.

## Results

**Overview of the approach.** Genomic prediction accuracy, defined here as the correlation coefficient between the phenotypes and estimated PRS, can be determined theoretically by heritability, $M_e$ and the sample size of reference data set[25,28,32] (see 'Methods'). The lower the $M_e$ value, the higher the prediction accuracy is. $M_e$ is a function of effective population size, and can be empirically estimated from the variance of genomic relationships between the reference and target samples[25,27] ('Methods').

In order to assess prediction accuracy, we used the UK Biobank data comprising 408,218 individuals after quality control. We identified 288,837 individuals that had no genomic relationships >0.05 with any of the other individuals in the data set (see 'Methods'). We randomly divided these unrelated individuals into discovery (80%) and target data sets (20%). We refer to this design as a 'large-scale design' (Fig. 1). Among all traits available to us, we chose 50 traits (Supplementary Table 1) with the highest heritability estimates according to estimates reported by the Neale lab[33]. These traits can be categorised as mental, physical and lifestyle traits. Trait name, type and SNP-based heritability estimated based on various information sources are shown in Supplementary Table 1. It is noted that narrow-sense heritability estimates using the unrelated individuals in the large-scale design generally agree with those from the Neale lab[33] (Supplementary Table 1).

In the large-scale design, we introduced the first-, second- or third-degree relatives (Fig. 1; Supplementary Table 2) that were identified from the total sample (408,218 individuals) according to their kinship coefficients inferred from genotype information. To perform a fair comparison with the analysis utilising the unrelated sample, we substituted the same number of unrelated individuals with the first-, second- or third-degree relatives such that the same sample size (i.e., 288,837 individuals) was consistently used across the genomic prediction analyses with various degrees of relationships. Thus, there were four different analyses with (1) unrelated sample only, (2) inclusion of first-, (3) second- and (4) third-degree relatives. It was noted that the number of substitutions in the analyses with first-, second- and third-degree relatives in the large-scale design varied between traits, depending on the number of available records for each trait in the large-scale design (Supplementary Table 3). The number of individuals in each level of relatedness for each trait is also reported in Supplementary Table 3.

In contrast to the large-scale design, we also evaluated a 'small-scale design' to quantify how the prediction accuracy was affected by the sample size and the proportion of high relationships. We selected 6000 individuals in each of first-, second- and third-degree relatives (see Fig. 1), using a greedy algorithm[34] that allowed to maximise overall relationships among the selected individuals, hence minimising $M_e$ (see 'Methods'). The proportion of relatives, hence the variance of relationship, was thereby increased in the small-scale design, compared to that in the large-

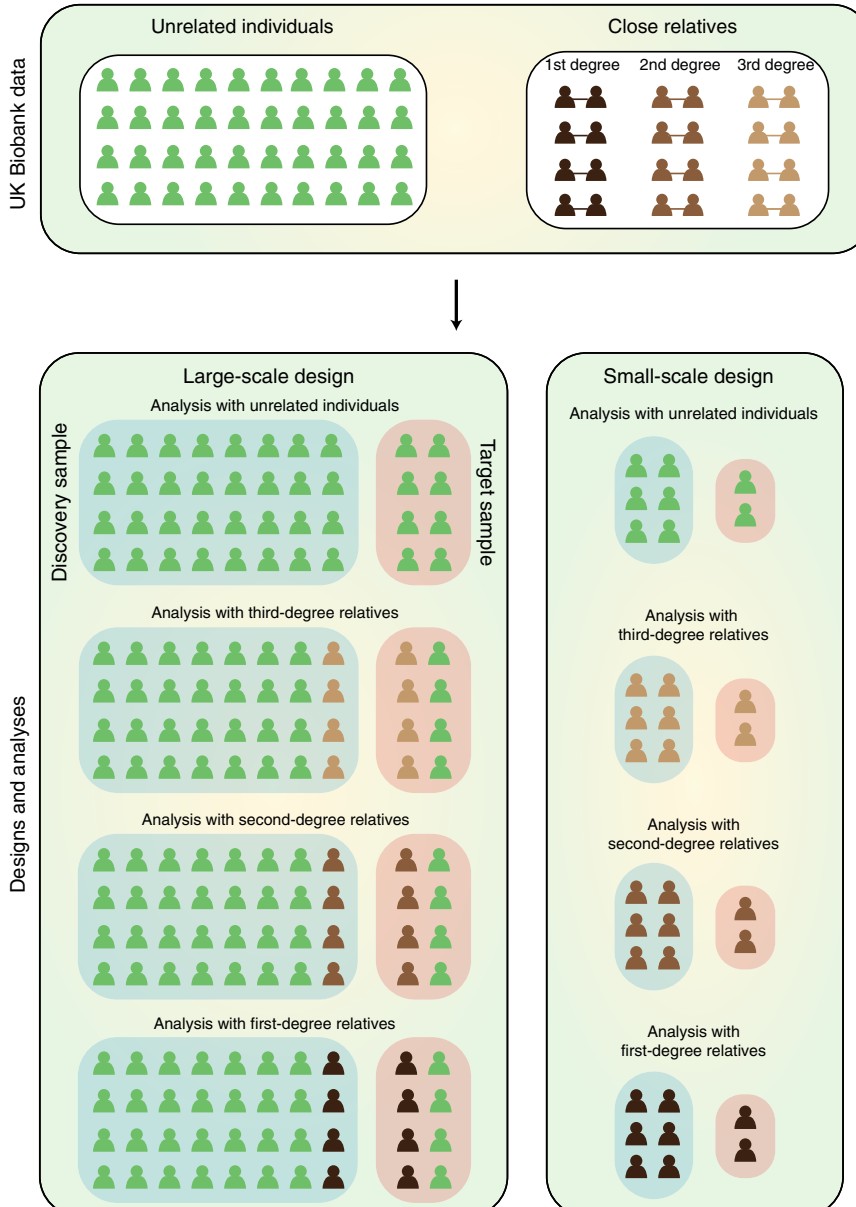

**Fig. 1 A schematic illustration for study designs and analyses.** We made large- and small-scale designs, each having four analyses with unrelated, first-, second- and third-degree relationships between discovery and target samples. Initially, we identified 288,837 unrelated individuals for whom any pairwise relationship was less than 0.05 (green). We also identified first- (dark brown), second- (brown) and third-degree (light brown) relatives using the information of kinship coefficients from the full UK Biobank sample. The analysis with the unrelated sample in the large-scale design was carried out for all unrelated individuals with available phenotypic information who were randomly divided into discovery (80%) and target data sets (20%). For the analysis with first-, second- or third-degree relatives in the large-scale design, the set of first-, second- or third-degree relatives were substituted with a set of the same number of randomly selected individuals in the analysis with unrelated sample. For the analysis with unrelated sample in the small-scale design, we used 6000 individuals (5000 as discovery, and 1000 as target sample), randomly selected from the analysis with unrelated sample in the large-scale design. However, with the analysis with first-, second- or third-degree relatives in the small-scale design, we selected 6000 individuals from the set of first-, second- or third-degree relatives, maximising the relationships among the selected individuals (see 'Methods').

scale design (Supplementary Table 3), as expected. As in the large-scale design, there were four different analyses in the small-scale design of 6000 individuals with (1) the unrelated individuals only, (2) the first-degree relationship pairs, (3) the second-degree relationship pairs and (4) the third-degree relationship pairs (see Fig. 1 and Supplementary Table 3).

In the target data set, prediction accuracy was empirically calculated as the correlation between the polygenic scores based on SNP effects estimated in the discovery data set[35] and the phenotypes adjusted for potential confounders (see 'Methods').

We also estimated $M_e$ and heritability using Eq. (2) ('Methods'), and further computed theoretical prediction accuracy using Eq. (1) ('Methods'), which were used to evaluate the empirical prediction accuracy of the designs. In this study, we defined estimated heritability based on unrelated individuals as narrow-sense heritability and estimated heritability based on familial relationships as family-based heritability[36] (see 'Methods').

Finally, we show how to utilise ungenotyped individuals to increase prediction accuracy further. Ungenotyped siblings of a target individual may have phenotypic information for the trait of

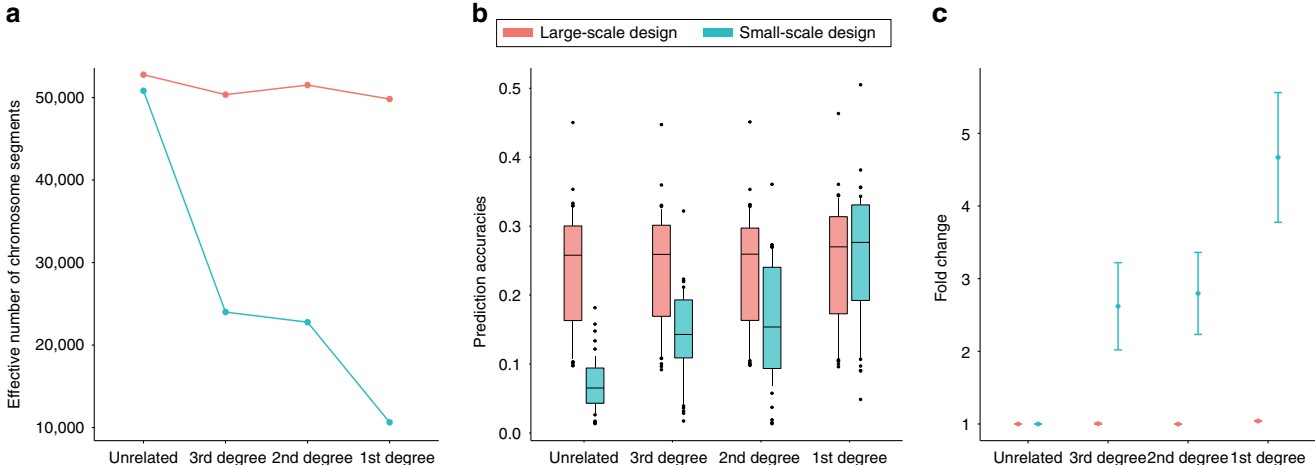

**Fig. 2 A decreased effective number of chromosome segments can improve the polygenic prediction accuracy.** The effective number of chromosome segments ($M_e$) (**a**), the actual prediction accuracy (**b**) and the fold change of the prediction accuracy from each degree relatives with respect to that from unrelated samples in the large- and small-scale design (**c**). Accuracy of polygenic scores was calculated as the correlation between the polygenic score and the phenotype adjusted for batch, assessment center, sex, age and ten principal components of ancestry. $M_e$ is computed by the inverse of variance of genomic relationships between discovery and target sample. The dot points and error bars in (**b**) and (**c**) represent the mean values and 95% confidence intervals from the analyses of 50 complex traits. The boxplots (**b**) show the first to the third quartile of prediction accuracies for 50 complex traits and the whiskers reflect the maximum and minimum values within 1.5 × interquartile range for each group.

interest that is useful to predict the target individual. In practice, the known pedigree relationships between a genotyped target individual and ungenotyped relatives of the target individual can be used to construct a (inferred) realised relationship matrix for all individuals, including ungenotyped relatives (see 'Methods'). This realised relationship matrix is named as **H**-matrix[37] ('Methods') that include the relationships between genotypes and ungenotyped individuals as well. We fit the **H**-matrix in a linear mixed model to obtain polygenic risk scores for the target individual using BLUP approach[13,38], which is referred to as HBLUP[14,38–40]. We compare the prediction performance of HBLUP and GBLUP that is based on genotyped individuals only and analogous to PRS approach.

**Improved polygenic prediction accuracy with decreased $M_e$.** In the large-scale design, prediction with close relatives was not significantly better than that with unrelated individuals only (Fig. 2; Supplementary Table 4). This was probably due to the fact that the effective number of chromosome segments was not much different between using the analysis with close relatives or unrelated individuals in the large-scale design (Fig. 2a). The negligible difference of $M_e$ is not surprising because the number of substituted individuals for the analyses with close relatives in the large-scale design was small (Fig. 1; Supplementary Table 2) such that the majority of individuals in each analysis with close relatives were still unrelated.

In the small-scale design, we observed significant fold changes of 2.62 (95% CI: 2.02–3.22, $P$-value from a two-tailed paired $t$ test = 2.86E-06), 2.80 (95% CI: 2.23–3.36, $P$-value = 1.00E-07) and 4.67 (95% CI: 3.78–5.56, $P$-value = 1.59E-10) when comparing the prediction accuracy of the analyses with third-, second- and first-degree relatives, respectively, to that of the analysis with unrelated individuals only (Fig. 2c; Supplementary Table 4). This significant improvement of prediction accuracy can be explained by a dramatic decrease of $M_e$ for each analysis with close relatives, compared to that with unrelated sample only in the small-scale design (Fig. 2a; Supplementary Table 5). Thus, the contrasting results between the large and small-scale designs can be explained by substantially larger proportion of close relatives in the small-scale design than in the large-scale design (Supplementary

Table 2). Note that a small difference between $M_e$ values from the analyses with 2nd and 3rd degree relatives in the small-scale design was because the number of 2nd degree relatives was substantially less than the number of the 3rd degree relatives (see 'Methods').

We also used 5000 discovery samples and two sets of target data sets, each with 1000 target samples that were related ($T_A$) or unrelated ($T_B$) to the reference samples, which were considered in the same prediction analysis for a fair comparison (Supplementary Fig. 1). It was shown that the prediction accuracy for $T_A$ was much higher than $T_B$, confirming the results depicted in Fig. 2b. The low prediction accuracy for $T_B$ was because of the fact that the increase in accuracy is limited to the samples in the target set that do have close relatives in the discovery set, as expected from theory.

When reference sample size increased from 5000 to 10,000 or 15,000 for the analysis with first-degree relatives, the prediction accuracy increased further (Supplementary Table 4). Compared to using unrelated individuals in the large-scale design, the prediction accuracy with 10,000 or 15,000 reference individuals including first-degree relatives was significantly higher than that with 220,000 unrelated individuals (fold change = 1.17, 95% CI: 1.10–1.25, $P$-value from a two-tailed paired $t$ test = 4.12E-05, or 1.18, 95% CI: 1.11–1.25, $P$-value = 1.78E-05).

Supplementary Fig. 2 illustrates analytically how the $M_e$ value of a single-target individual changes when adding his or her close relatives to the reference data, given our study design. For example, in the small-scale design, the $M_e$ value decreases from 50,000 to 10,000 when adding 2 or 3 full sibs (i.e., first-degree relatives) of the target individuals in the reference data. When adding 2nd or 3rd degree relatives, a higher number of relatives are required to obtain the same $M_e$ value (Supplementary Fig. 2a). Given Eq. (1), it is likely that the prediction accuracy increases with lower $M_e$ values (Supplementary Fig. 2b), which clearly agrees with the observed empirical accuracy (Fig. 2).

**Empirical prediction accuracy compared with theoretical prediction accuracy.** Because we used information from close relatives, the empirical accuracy would be influenced by familial environmental effects that are not accounted for when estimating

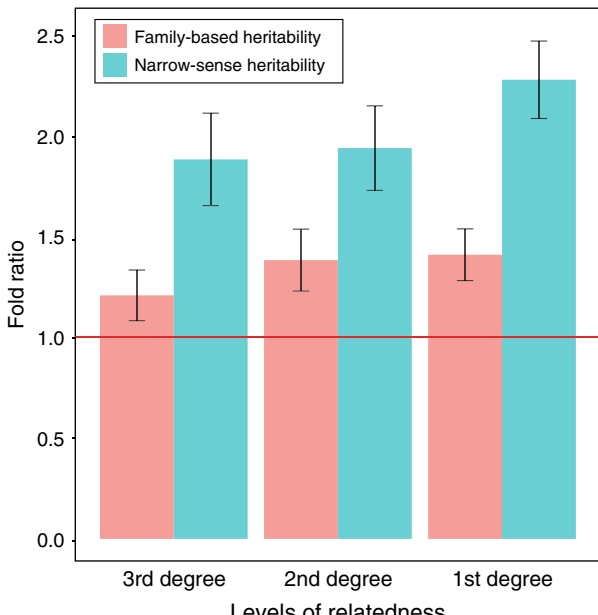

**Fig. 3 The ratio between the empirical and theoretical prediction accuracies in the small-scale design.** The theoretical prediction accuracy was calculated with family-based heritability or narrow-sense heritability. Narrow-sense heritability is pre-computed by Neale lab[33] from the original UKB data with unrelated individuals. Family-based heritability of each trait is estimated for each trait by GREML using small-scale design[10]. Theoretical prediction accuracy is formulated as a function of the effective number of chromosome segments, heritability and the number of phenotypic observations for each trait. The effective number of chromosome segments is estimated as the inverse of the variance of relationships between reference and target sample. The main bars represent the mean values averaged over the analyses of 50 traits. The error bars show the 95% confidence intervals of the mean values. The red horizontal line indicates a ratio of 1.

theoretical accuracy (Eq. (1) in 'Methods'). In order to quantify the familial environmental effects, we compared the empirical and theoretical accuracy for the small-scale design (Fig. 3; Supplementary Fig. 3), showing that the difference between the empirical and theoretical accuracy was proportional to the degree of relatedness. Note that we used estimates of both narrow-sense and family-based heritabilities (see 'Methods') when obtaining theoretical prediction accuracy (Supplementary Table 6).

When using narrow-sense heritability reported from the Neale lab[33], there were 1.88 (95% CI: 1.65–2.11, P-value from a two-tailed paired t test = 9.58E-10), 1.94 (95% CI: 1.73–2.15, P-value = 1.225E-11) and 2.28-fold change (95% CI: 2.09–2.47, P-value < 2.2E-16) in the comparison of the empirical prediction accuracy to the theoretical prediction accuracy for the analyses with third-, second- and first-degree relatives, respectively, in the small-scale design (Fig. 3). However, when using family-based heritability estimated from the small-scale design, the difference between the empirical and theoretical prediction accuracies reduced significantly although the fold changes were still deviated from 1, that is, 1.21 (95% CI: 1.08–1.34, P-value = 0.002), 1.38 (95% CI: 1.23–1.54, P-value = 1.1E-05) and 1.41 fold change (95% CI: 1.28–1.54, P-value = 9.205E-08) for third-, second- and first-degree relatives, respectively (Fig. 3).

**Efficient polygenic risk prediction using small-scale design with relatives.** It is well known that the prediction accuracy increases when using a larger sample size[21,23,24]. Here, we

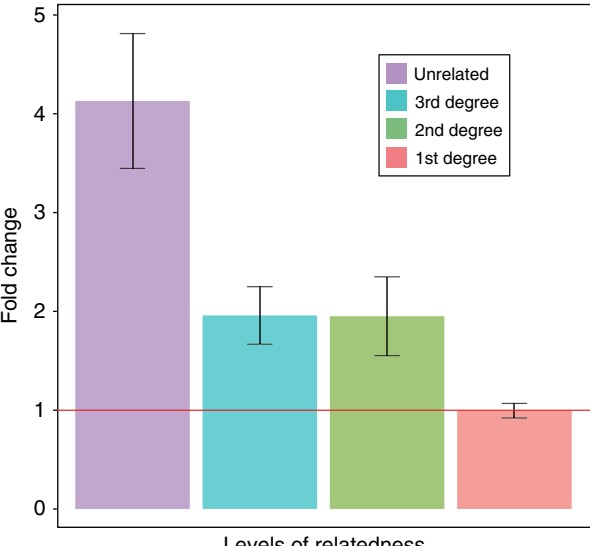

**Fig. 4 The ratio of the empirical prediction accuracies between large- and small-scale design.** The empirical prediction accuracy of the large-scale design is compared to that of the small-scale design for each level of relatedness. The main bars represent the mean values of ratios averaged over the analyses of 50 traits. The error bars show the 95% confidence intervals of the mean ratios. The red horizontal line indicates a ratio of 1.

investigated how the prediction performance was affected by integrating information from close relatives. When using unrelated sample only, the large-scale design performed better than the small-scale design (4.13-fold change, CI 95%: 3.45–4.82, P-value from a two-tailed paired t test = 5.99E-12), as expected. However, when using analyses with close relatives, the difference between the large and small-scale design became negligible, i.e., no difference for the analysis with first-degree relatives (mean = 1.00-fold change, 95% CI: 0.92–1.07, P-value = 0.905) (Fig. 4) although the difference in sample size is 44-fold. Notably, the empirical prediction accuracy with the average discovery sample size of ~220,000 unrelated individuals was not better than that with the 5000 individuals with first-degree relationships (mean = 0.96-fold ratio, 95% CI: 0.88–1.03, P-value = 0.289; Supplementary Fig. 4).

Next, we classified the traits into three types, namely mental, physical and lifestyle traits (see 'Methods' and Supplementary Table 1). With unrelated individuals only, the prediction accuracy using the large-scale design was significantly higher than that using the small-scale design for all types of traits (Fig. 5a). However, for the analyses with first-degree relatives, the performance of PRS with the large-scale design was not significantly different from that with the small-scale design in mental or physical traits (Fig. 5b). For lifestyle traits, the prediction accuracy of the small-scale design with first-degree relatives was even significantly higher than that of the large-scale design (mean prediction accuracy = 0.22 vs. 0.16, fold change = 1.40, 95% CI = 1.17–1.62, P-value from a two-tailed paired t test = 0.025) (Fig. 5b), which is remarkable. For analyses with 2nd and 3rd degree relatives, we found that PRS using a large sample size would be more predictive of phenotypes, compared to using a small sample size although the difference was marginal in general (Supplementary Fig. 5).

**Clinical impact of polygenic risk scores when using close relatives.** Given the efficient predictive performance from the small-scale design with first-degree relatives, we evaluated the prevalence and odds ratio in the decile analyses of PRS using 12

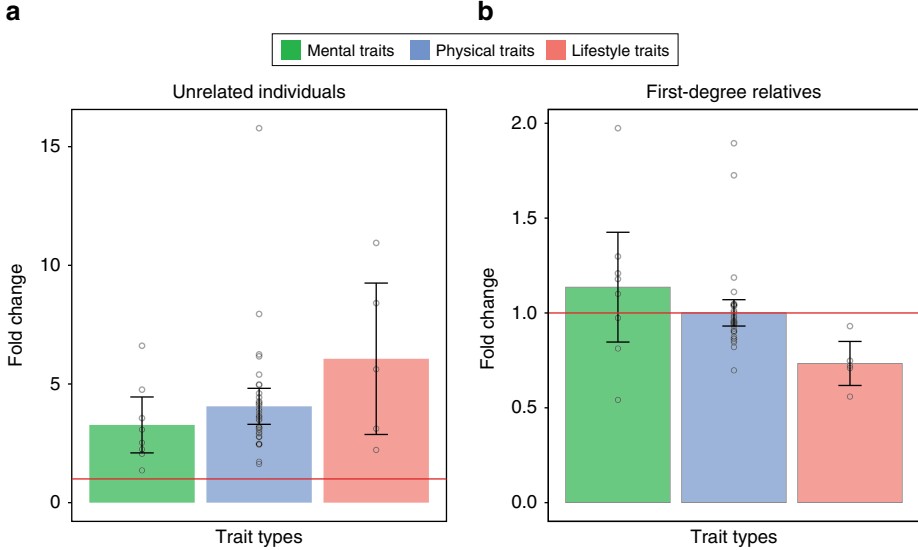

**Fig. 5 The ratio between the empirical prediction accuracies in the large- and small-scale designs for three types of traits.** The fifty traits were classified into three types of traits, i.e., mental, physical and lifestyle traits, for the analyses with unrelated individuals (**a**) and first-degree relatives (**b**). The main bars represent the mean values of ratios averaged over the analyses of 8, 37 and 5 traits in mental, physical and lifestyle traits, respectively. The error bars show the 95% confidence intervals of the mean ratios. The red horizontal line indicates a ratio of 1.

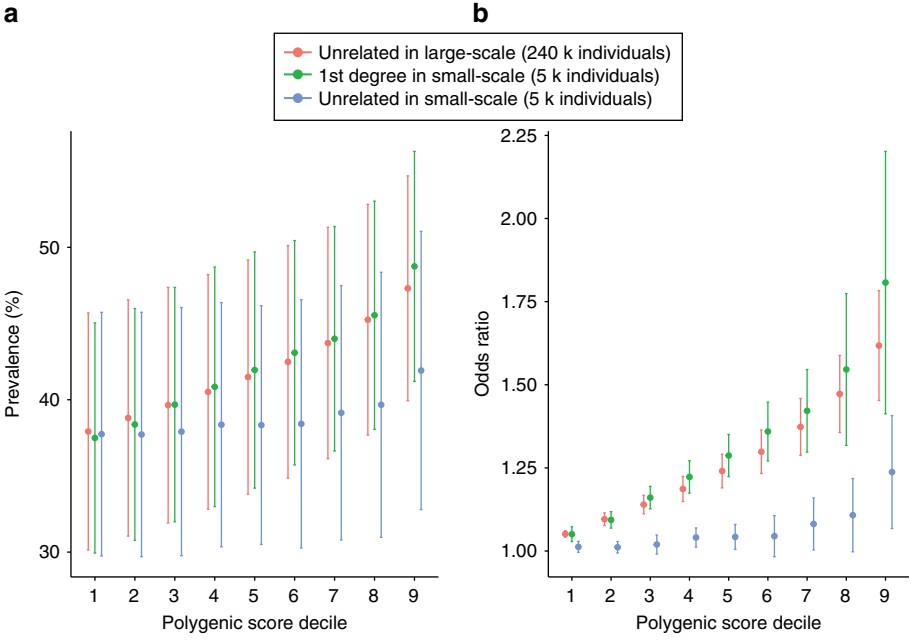

**Fig. 6 Clinical impact of polygenic risk scores when using dichotomous traits.** Prevalence of cases (**a**) and odds ratio (**b**) in decile analyses of polygenic risk scores for 12 dichotomous traits are shown. For each trait, the target individuals were divided into ten deciles according to their PRS (1 = lowest and 10 = highest). The prevalence was calculated as the proportion of cases in the target individuals above each decile. The odds ratios were calculated from the odds (case/controls) for the target individuals above each decile that was divided by the odds for the whole UKB individuals (i.e., general population). The dot points and error bars represent the mean value and 95% confidence interval over the analyses of 12 traits.

binary traits selected from the 50 complex traits (Supplementary Table 1). The prevalence in the top PRS above the 1st–9th decile of PRS varied and increased on average from 37.9% to 47.3% in the analysis with unrelated individuals in the large-scale design. Similarly, in the analysis with first-degree relatives in the small-scale design, the prevalence increased from 38% to 48.1%. However, the prevalence of these dichotomous traits with unrelated individuals from the small-scale design was low (a prevalence of 41.9% even for the top 10% PRS) (Fig. 6a). The ratio of case–control odds ratio for the top decile against the whole UK

Biobank population was 1.62 (95% CI: 1.45–1.78, *P*-value from a two-tailed paired *t* test = 1.48E-05) and 1.81 (95% CI: 1.41–2.20, *P*-value = 0.002) when using unrelated individuals in the large-scale design and first-degree relatives in the small-scale design, respectively (Fig. 6b). On the other hand, PRS using unrelated individuals in the small-scale design had negligible power to contrast the top decile and the whole population (Fig. 6b). The large-scale design (using unrelated sample) could reach an odds ratio of 4.08 (95% CI: 2.95–5.21, *P*-value = 0.0002) in the top 0.05% of PRS (Supplementary Fig. 6). Due to the limited sample

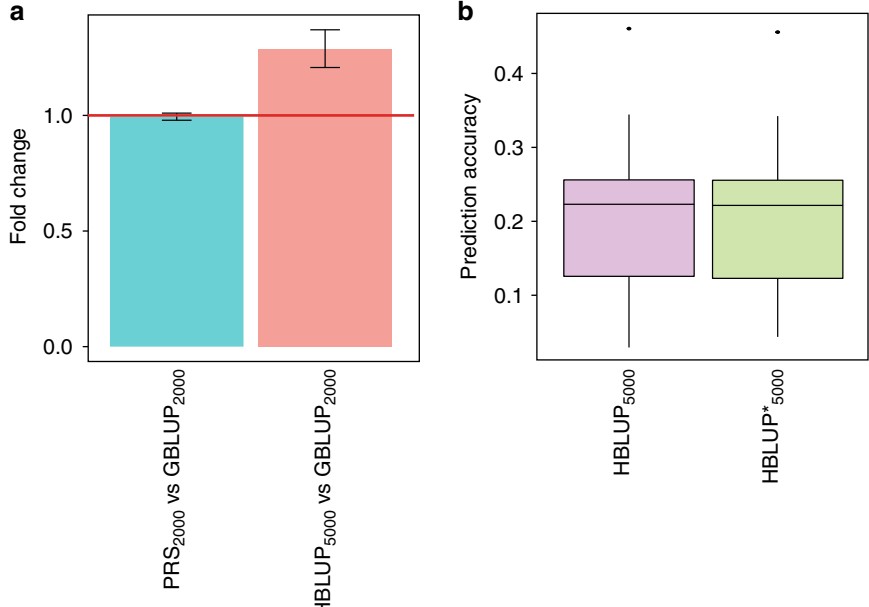

**Fig. 7 Prediction performances with and without phenotypic information of ungenotyped relatives of the target sample. a** The fold change of prediction accuracy using PRS from GWAS summary statistics with respect to that using GBLUP (PRS$_{2000}$ vs GBLUP$_{2000}$) and the fold change of HBLUP accuracy with respect to GBLUP accuracy (HBLUP$_{5000}$ vs GBLUP$_{2000}$). The main bars represent the mean values of the fold changes averaged over the analyses of 50 complex traits. The error bars show the 95% confidence intervals of the mean fold changes. The red horizontal line indicates a ratio of 1. The subscript in the name of each method represents the sample size in the discovery data set. **b** HBLUP prediction accuracy with (HBLUP$_{5000}$) and without (HBLUP*$_{5000}$) using pedigree information between ungenotyped relatives of the target samples and other individuals in the discovery sample. The boxplots show the first to the third quartile of prediction accuracies for 50 complex traits, and the whiskers reflect the maximum and minimum values within 1.5 × interquartile range for each group. PRS$_{2000}$: Polygenic Risk Score with 2000 genotyped individuals only, GBLUP$_{2000}$: best linear unbiased prediction integrated with genomic relationship matrix with 2000 genotyped individuals only. HBLUP$_{5000}$: best linear unbiased prediction with H-matrix including 2000 genotyped and 3000 ungenotyped relatives of the target samples in the discovery sample.

size in the small-scale design, we could not compare the performance in the extreme percentile groups. Detailed prevalence and odds ratio for each dichotomous trait are provided in Supplementary Table 7 and Supplementary Table 8.

**Further improvement of prediction accuracy using ungenotyped relatives.** Ungenotyped relatives of target sample have potential to contribute to predicting future phenotypes[15,30,41]. For the analysis with first-degree relatives in the small-scale design ($n = 6000$), we assumed that only a random half of the sample was genotyped, and the other half was not genotyped. Among the genotyped individuals ($n = 3000$), 2000 and 1000 individuals were used as discovery and target sample, respectively, in the prediction using the PRS approach or GBLUP (see 'Methods'). We also used HBLUP that could additionally utilise the information from the ungenotyped relatives ($n = 3000$) of genotyped individuals ($n = 3000$). The prediction performances across the methods, PRS, GBLUP and HBLUP, were compared. Figure 7a showed that the prediction accuracy with 2000 remaining individuals was invariant whether using PRS or GBLUP (0.99-fold change, 95% CI: 0.98–1.01, P-value from a two-tailed paired $t$ test = 0.48). When including ungenotyped individuals, HBLUP outperformed other methods. For instance, with 3000 ungenotyped individuals, the prediction accuracy achieved by HBLUP with additional ungenotyped individuals was better than GBLUP with only 2000 genotyped individuals (1.289-fold change, 95% CI: 1.207–1.37, P-value = 8.23E-09) (Fig. 7a). The prediction accuracy was positively correlated with the number of ungenotyped relatives (i.e., sample size) and heritability (Supplementary Fig. 7 and Supplementary Table 9). As expected, the best prediction performance could be achieved when all individuals were

genotyped (Supplementary Fig. 7). The values of prediction accuracies across various methods can be found in Supplementary Table 10 and Supplementary Fig. 8. It is noted that the prediction accuracy of HBLUP was invariant whether or not using pedigree information between ungenotyped relatives and the other discovery sample (Fig. 7b).

## Discussion

We demonstrated that the polygenic prediction utilising close relatives between reference and target samples outperformed the analyses with unrelated individuals only by using the small-scale design. Compared with the analyses with second- or third-degree relatives, or unrelated individuals, a higher prediction accuracy was observed from the analysis with first-degree relatives, which was because of a lower value of $M_e$ that required fewer independent parameters to be estimated[25–27]. Moreover, this higher prediction accuracy was also probably due to the fact that close relatives share some unknown (unmodeled) factors in addition to additive genetic effects, which may be dominance, gene-by-family interaction and familial environmental effects. It was also shown that the analyses with second- and third-degree relatives outperformed the analysis with unrelated individuals although they were less efficient to improve the prediction accuracy, compared to first-degree relatives.

The approach of including close relatives will be most useful in applications where accuracy matters more than delineating between causal genetic effects and other effects. It is known that family-based heritability estimates can be inflated if nonadditive genetic effects or common environmental effects shared between close relatives are confounded with additive genetic effects[3], which can be considered biased according to the concept of

narrow-sense heritability that includes the additive genetic effects only. However, this bias should not be an issue when predicting the future phenotypes of target sample (i.e., a new-born baby) because such nonadditive genetic and common environmental effects can be a valuable source to improve the prediction accuracy[28,42]. Indeed, family history has been widely used as a biomarker to predict disease risk[43,44], and it can also be used to increase the power to identify causal variants in GWAS[45–47]. We consider that our method is a more systematic approach to utilise information of family history as well as within-family segregation[48].

The prediction performance with close relatives varied, depending on traits. For example, the prediction accuracy from the analysis with first-degree relatives in the small-scale design (a discovery sample size of 5000) was significantly higher than the prediction accuracy with unrelated individuals in the large-scale design (a discovery sample size of 220,000) for lifestyle (behavioural) traits such as drinking, smoking and qualification. However, this was not observed in mental or physical traits. This observation agrees with a previous study showing that educational achievement is more similar in dizygotic twins, compared with a mental trait such as neuroticism scores[49]. This suggests that polygenic prediction should be based on information from close relatives particularly for lifestyle and behavioural traits.

Previous studies reported the potential of PRS in clinical practice[7,50,51]. For instance, Khera et al. reported that the top 2.5% (high-risk group) identified by PRS was at fourfold changed risk compared with the remaining 97.5% for coronary artery disease, which is a similar predictive power when comparing carriers and non-carriers of a rare monogenic mutation associated with increased cholesterol[7]. Our finding emphasises on an implication of using close relatives that can increase the prediction accuracy substantially, compared with the existing PRS approaches. In the near future, it is likely that more close relatives can be genotyped, and this information should be efficiently used in clinical care.

We investigated if the predictive power increased when using ungenotyped relatives of target individual in polygenic risk prediction. Utilising information of ungenotyped relatives has been widely used in the genomic prediction of economic traits in other species, such as cattle[15,37,41]. However, to our knowledge, this has never been verified in human population studies in the context of polygenic risk prediction. Here, we explicitly verified the approach that could enhance the predictive power, using a large-scale human biobank data. We show that phenotypic information of ungenotyped relatives can be useful in polygenic risk prediction, which may have important implications in clinical practice.

There are a number of limitations in this study. A potential caveat of our analysis is the limited number of relatives in the data (i.e., only less than three close relatives for each target individual on average). This limitation obscured the actual predictive power as the number of relatives of each target individual should have been more than that from the UKB data that only include genotyped samples. It may be possible to trace the information of relatives of the genotyped individuals in the UKB data, and HBLUP can be used to integrate the information even though relatives do not have genotypic information, which is, however, beyond the scope of this study. We anticipate that the number of relatives will increase as the scale of biobank data increases. Another limitation is that the number of lifestyle traits is only four and a further study may be required to confirm the finding about the prediction of lifestyle traits. It is also noteworthy that our study focused on individuals with European ancestry only. More studies on other ethnicities will be desirable. This caveat has been recently raised when a PRS application comes to clinical practice[52]. Thirdly, although it is well

established how to obtain the theoretical prediction accuracy based on genotypic infromation[25,32,53], or based on pedigree information (e.g., using selection index theory)[54], there is no unified theoretical approach to derive the expected prediction accuracy that can be applied to combined genotyped and ungenotyped samples, i.e., in HBLUP framework. Lastly, HBLUP is computationally demanding, which prevents using the ungenotyped relatives of target individuals in a large-scale data. It is required to develop an efficient HBLUP method, i.e., based on summary statistics.

Polygenic risk scores based on genome-wide SNP information will provide useful information to predict the future phenotypes of target individual, which allows an early prevention of complex diseases. The cost of genome-wide genotyping has been dramatically reduced in the last decade, and multiple genotyping services are publicly available. In fact, genomic databases such as biobank datasets (e.g., UK, All of us, Estonian, Japanese and Chinese-Kadoori)[52] and commercial genotyping databases (23andMe, Ancestry and MyHeritage)[55] have clinical measures or can be linked with existing national clinical databases with relative information available. Moreover, prenatal genetic tests with information from close relatives have shown a prospective to provide insights for several phenotypes[56,57]. In the near future, it is likely that there is a high probability of finding genotyped close relatives of a random sample[58], and the prediction of their (future) phenotypes benefits from the information of already known genotypes and phenotypes of their relatives. Here, we show how to use the information of relatives and highlight the importance of their phenotypes and genotypes in polygenic risk prediction. Our findings will have a useful implication for future investigations into precision health and preventive medicine.

## Methods

UK Biobank's scientific protocol has been reviewed and approved by the North West Multi-centre Research Ethics Committee (MREC), National Information Governance Board for Health & Social Care (NIGB), and Community Health Index Advisory Group (CHIAG). UK Biobank has obtained informed consent from all participants. Research Ethics approval was obtained from University of South Australia Human Research Ethics Committee (HREC).

**Data and quality control**. The UK Biobank (UKB) enrolled 488,377 individuals and 92,693,895 imputed SNPs across autosomes[59]. For each individual, a trained nurse or an automatic device undertook a series of anthropometric measurements and surveys. In our study, a stringent quality control protocol was applied. SNPs were excluded according to the following criteria: INFO score < 0.6, MAF < 0.01, Hardy–Weinberg Equilibrium $P$-value < 1E-7 and missingness > 5%, one SNP was randomly chosen to keep if there is duplicated SNPs. We only used SNPs from HapMap 3 due to their reliability and robustness to bias in the estimation of narrow-sense heritability. In terms of individuals filtering, individuals with a genotype calling rate <0.95 were excluded. We performed analyses on the samples of white British only. These filters remained 1,133,273 SNPs and 408,218 individuals. In addition, we removed ambiguous or duplicated SNPs. We calculated the discordance rate between imputed genotypes of the first and the second release of UKB data, and individuals and SNPs with a discordance rate larger than 0.05 were removed. Moreover, we excluded individuals whose first- or second-principal components exceeded 6 standard deviations from the population mean (white British). We also randomly excluded one individual from any pair of related individuals with a genomic relationship larger than 0.05. After these QC steps, 288,837 unrelated individuals and 1,130,918 SNPs remained.

We analysed 50 complex traits (see Supplementary Table 1) without missing information in individual data and have the highest SNP-based heritability estimates, which were significantly different from zero ($P$-values from a Wald test <0.05) reported by Neale lab[33]. The number of individuals with available information ranged from the lowest number of 237,191 individuals with heel bone mineral density (BMD) T-score (automated) (UKB ID = 78) to the highest number of 407,938 individuals with alcohol intake frequency (UKB ID = 1558). These 50 traits could be categorised into mental, physical and lifestyle traits.

For a continuous trait with multiple response items, we calculated the averaged value of multiple responses, i.e., diastolic/systolic blood pressure and pulse rate. For categorical traits, e.g., Qualifications: College or University degree, individuals with College or University degree were marked as cases and the remaining individuals are marked as controls.

**Large- and small-scale design for risk prediction**. In the large-scale design, we used four different analyses with (1) unrelated individuals only, (2) inclusion of first-, (3) second- and 4) third-degree relatives according to their kinship coefficients. In each analysis with relatives, we substituted the same number of unrelated individuals with the first-, second- or third-degree relatives available in the data. The kinship coefficients and (genomic) relationships used to classify the analyses in the large-scale design were derived in the following process.

Genomic relationship matrix is computed by PLINK version 1.9[60]. Level of relatedness is determined by kinship coefficient which is defined as the probability that a pair of randomly homologous alleles are identical by descent. Kinship coefficient is inferred by KING software version 2.1[61]. Relatedness thresholds and the proportion of close relatives substituted in the sample of unrelated individuals for the analyses of first-, second- and third-degree relatives are described in Supplementary Table 2.

With 288,837 unrelated individuals of pairwise genomic relationship < 0.05, we identified 279,020 individuals with available phenotypic information on average. For the large-scale design, we partitioned these unrelated individuals into a proportion of 80% (223,215 individuals) and 20% (55,803 individuals) for discovery and target samples, respectively. To compare with analysis with unrelated sample, we replaced unrelated pairs with close relatives identified by kinship coefficient. We introduced the first-, second- or third-degree relatives (Fig. 1; Supplementary Table 2) that were identified from the total sample (408,218 individuals) according to their kinship coefficients inferred from genotype information. We removed duplicated individuals within and between discovery and target sample to avoid bias in the analyses.

In the small-scale design, we used four different analyses with (1) unrelated individuals only, (2) first-, (3) second- and (4) third-degree relatives, in a similar manner as in the large-scale design, but with a small sample size. For the analysis with unrelated individuals only, we randomly selected 6000 individuals from 279,020 unrelated individuals. For the analyses with each level of relatedness, we used a graph and network analysis tool[62] to maximise the average relatedness among selected individuals from the set of each level of relatedness. For example, for the first-degree relatives, each individual, who has one or more of first-degree relatives, is represented as a node and their first-degree relatives are linked through undirected edges, using igraph version 1.2.5 package[62]. The number of individuals in each group varied from 2 to 6 members. We then selected groups with the highest number of individuals starting from groups with six members to groups with two members until we achieved 6000 individuals. Based on the selected individuals, we randomly assigned 5000 individuals into the discovery data set and the remained 1000 individuals were used as the target sample. These steps were equally applied to second- and third-degree relatives. The average numbers of individuals per family for each level of relatedness is reported in Supplementary Table 11.

To compare prediction accuracies between analyses, for instance, using first-degree close relatives in the large-scale design against small-scale design, we computed the mean fold change across a variety of different traits with its 95% confidence interval and assessed the statistical significance level whether the fold change was significantly different from 1 with a two-tailed paired $t$ test.

**Estimation of Polygenic Scores in UK Biobank sample**. *Polygenic Risk Score (PRS)*: The phenotypes of each trait were adjusted for batch information, centre, sex, age and population stratification (using the first ten principal components), using a linear regression. The pre-adjusted phenotypes were used for the following GWAS and PRS analyses. We estimated SNP effects by conducting GWAS for each of 50 traits using the discovery sample of 223,215 and 5000 individuals in the large-scale and small-scale design, respectively. PRS were calculated for the target individuals (55,803 and 1000 individuals for large-scale and small-scale sample, respectively), as the sum of the risk alleles weighted by the estimated SNP effects from the GWAS using the discovery sample only. Then, we obtained the correlation between the PRS and pre-adjusted phenotypes in the target data set. For these analyses, we used PLINK version 1.9[60] and PRS were computed using PRSice version 2.1.11[35].

*Genomic best linear unbiased prediction (GBLUP)*: We used GBLUP[14,38–40,63,64] to generate polygenic score for each individual utilising the genomic relationships between individuals. GBLUP fits a genomic relationship matrix that is estimated based on the 1,130,918 genome-wide SNPs, which can be written as

$$\mathbf{G} = \mathbf{WW}\prime/M,$$

where **G** is genomic relationship matrix, **W** is the matrix for individual genotypic information coded as 0, 1 or 2 and $M$ is the number of SNPs. This analysis is conducted by MTG2 version 2.15[65].

*A matrix-based best linear unbiased prediction using pedigree information (ABLUP)*: ABLUP can be used to estimate polygenic scores, fitting **A** matrix that is based on pedigree information only without genotypic information.

*Polygenic prediction for ungenotyped relatives (HBLUP)*: In the small-scale design, we randomly chose 3000 individuals in the discovery set to set as missing genotyped. We first reconstructed the pedigree from genotypic information by PRIMUS[66] version 1.9.0 to obtain pedigree. After reconstructing pedigree, we removed individuals with ambiguous information identified by PRIMUS. With the reconstructed pedigree, we computed genomic-pedigree relationship matrix (**H**-

matrix) from genomic relationship matrix (**G**) and numerator relationship matrix (**A**)[15,37,41]. **A** matrix was solely based on pedigree information. **H**-matrix is computed as follow[37,67]:

$$\mathbf{H} = \begin{bmatrix} \mathbf{H}_{11} & \mathbf{H}_{12} \\ \mathbf{H}_{21} & \mathbf{H}_{22} \end{bmatrix} = \begin{bmatrix} \mathbf{A}_{11} + \mathbf{A}_{12}\mathbf{A}_{22}^{-1}(\mathbf{G} - \mathbf{A}_{22})\mathbf{A}_{22}^{-1}\mathbf{A}_{21} & \mathbf{A}_{12}\mathbf{A}_{22}^{-1}\mathbf{G} \\ \mathbf{G}\mathbf{A}_{22}^{-1}\mathbf{A}_{21} & \mathbf{G} \end{bmatrix}.$$

Here, we use the subscript 1 for ungenotyped and 2 for genotyped individuals. This approach is also known as the single-step approach in livestock genetics[15,37]. We applied a BLUP approach to calculate polygenic scores fitting the **H**-matrix (HBLUP). All calculations were computed in MTG2 version 2.15[65].

**Theoretical genomic prediction**. The theoretical accuracy of genomic prediction can be derived, taking into account heritability ($h^2$), the number of effective chromosome segments ($M_e$) and the sample size in the reference data set ($N$)[25,32]. $M_e$ can be empirically estimated as the inverse of the variance of genomic relationships between the discovery and target sample[25,28]. In the large-scale design, we estimated $M_e$ from samples who were available for standing height trait (UKB ID 50), and used the estimated $M_e$ to obtain theoretical prediction accuracies for the 50 traits. This was because the empirical estimation of $M_e$ was computationally demanding and samples available for other traits were mostly overlapping and homogeneous with those for standing height trait. To obtain the theoretical accuracy of genomic prediction, we used Equation 1[25,28,32]. Pasanuic[68] and Dudbridge[53] et al. introduced a theoretical prediction accuracy when using random-effects model (i.e., GBLUP) although it is not substantially different from Eq. (1).

Previous studies[25,28,32] have shown theoretical genomic prediction accuracy for a trait, which can be formulated with

$$r_{y,\hat{g}} = h \times r_{g,\hat{g}} = h \times \sqrt{\frac{h^2}{h^2 + \frac{M_e}{N}}} = \frac{h^2}{\sqrt{h^2 + \frac{M_e}{N}}}, \qquad (1)$$

where $r_{y,\hat{g}}$ is the correlation coefficient between the true and estimated genetic scores, $h^2$ is the heritability of a trait, $M_e$ is the effective number of chromosome segments[25–27] and $N$ is the number of phenotypic observations. Equation (1) shows that $M_e$ plays a key role in the prediction performance in addition to $h^2$ and $N$. A smaller number of independent chromosome segments can be estimated more accurately with the same number of records. $M_e$ is a function of effective population size and can be empirically estimated as[25,27]

$$M_e = \frac{1}{var\left(\mathbf{G}_{ij}\right)}, \qquad (2)$$

where $\mathbf{G}_{ij}$ is the genomic relationship between individual $i$ in discovery and individual $j$ in the target sample[28]. It is expected from Eq. (2) that including high relationships in **G** (i.e., close relatives) reduces the values of $M_e$, hence increases the prediction accuracy.

**Analytically derived $M_e$ values for a single-target individual when adding its relatives**. When adding relatives of a single-target individual in the reference data set, the variance of genomic relationships between the target individual and reference sample is changed, hence $M_e$ value is also changed. We considered various numbers of relatives of a single-target individual in the discovery sample in both small- and large-scale design to analytically quantify how Me values were changed and to assess the prediction accuracy using Eq. (1). In the analytical derivation, we used the existing genomic relationships between the target individual and the discovery sample from the large- and small-scale designs, and added 0.125, 0.25 or 0.5 to the relationships when adding third-, second- or first-degree relative of the target individual.

**Heritability estimation**. Heritability can be estimated using unrelated individuals, whose covariances would be determined by the additive genetic effects only, therefore it is a narrow-sense heritability. On the other hand, family-based heritability[36] from a sample with familial relationships included both additive genetic effects and remaining family effects. We define two kinds of models and heritabilities as

$$y_j = \begin{cases} g_j + e_j, \\ g_j + f_j + e_j, \end{cases} \text{ and}$$

$$h^2 = \begin{cases} \sigma_g^2/\left(\sigma_g^2 + \sigma_e^2\right), & \text{for narrow} - \text{sense heritability (3)} \\ \left(\sigma_g^2 + \sigma_f^2\right)/\left(\sigma_g^2 + \sigma_f^2 + \sigma_e^2\right), & \text{for family} - \text{based heritability (4)} \end{cases}$$

where $y_j, g_j, f_j$ and $e_j$ is the phenotypic value, additive genetic effects, and familial effects, and residual effects for the $j$th individual, respectively. Similarly, $\sigma_g^2, \sigma_f^2$ and $\sigma_e^2$ are the variance of the genetic, family and residual effects, respectively. We used both narrow-sense and family-based heritabilities to compare empirical and theoretical prediction accuracies (Eq. (1)).

We used LDSC[16] to estimate narrow-sense heritability as it is appropriate to deal with a large number of individuals while estimating family-based heritability could be done by GREML[10] using MTG2 version 2.15[65] because of a small number of close relatives.

**Reporting summary**. Further information on research design is available in the Nature Research Reporting Summary linked to this article.

## Data availability
The raw genetic and phenotypic data used in this study are available from UK Biobank. The UK Biobank data are publicly accessible through the procedure described in the webpage, http://www.ukbiobank.ac.uk/using-the-resource/. The source code for MTG2 version 2.15 is publicly available in https://sites.google.com/site/honglee0707/mtg2. The source data underlying Figs. 1–7 and Supplementary Figs. 1–8 are provided as a Source Data file. All other intermediate data generated in the downstream analyses in this study are available upon request. Source data are provided with this paper.

## Code availability
Code reported in this paper is available from: for LDSC, see https://github.com/bulik/ldsc; for MTG2, see https://sites.google.com/site/honglee0707/mtg2; for UK Biobank, see http://www.ukbiobank.ac.uk/; for PLINK1.9, see https://www.cog-genomics.org/plink/1.9/; for PRIMUS, see https://primus.gs.washington.edu/primusweb/; for PRSice, see https://choishingwan.github.io/PRSice/; for igraph, see https://igraph.org/redirect.html. Source data are provided with this paper.

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

## Acknowledgements

We thank Peter M. Visscher for giving valuable comments and criticisms for this work. This research is supported by the Australian Research Council (DP190100766, FT160100229) and the NHMRC Grant (No: 1123042). This research has been conducted using the UK Biobank Resource. UK Biobank (http://www.ukbiobank.ac.uk) Research Ethics Committee (REC) approval number is 11/NW/0382. Our reference number approved by UK Biobank is 14575. The funding body had no role in the design of the study, the collection, analysis and interpretation of the data and in writing the paper. This work was supported by computational resources provided by the Australian Government through NCI: Raijin under the National Computational Merit Allocation Scheme.

## Author contributions

S.H.L., B.T. and T.D.L. conceived the idea. S.H.L. and T.D.L. directed and supervised the study. B.T. performed the analyses. X.Z. and J.S. collected the data and conducted quality control. J.L. and J.v.d.W. provided critical feedback and key elements in interpreting the results. B.T. and S.H.L. drafted the first paper. All authors contributed to editing and approval of the final paper.

## Competing interests

The authors declare no competing interests.

## Additional information

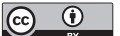

