## [Peer Review File · Nature Communications]

Reviewers' comments:

Reviewer #1 (Remarks to the Author):

This is a useful paper that describes a design in which related individuals are used in the reference panel when making polygenic risk score prediction. The UK Biobank dataset is used to explore the utility of this design. A few findings are reported. First, in a small scale design (6,000 total individuals), the prediction accuracy increases when more related individuals are included between the reference and target sets. However, this improvement is not seen in a large scale design where the related individuals only constitute a small fraction. Second, the empirical accuracy seems to agree with the theoretical prediction based on family based heritability, suggesting that familial environmental effect is small. Third, PRS prediction can be achieved the same accuracy with relatives even with small sample size. Fourth, the authors also demonstrated how this could be used in a clinical setting using binary traits. Finally, the prediction can be further improved with ungenotyped relatives.

Overall, the paper is easy to follow. However, I find nothing new in this paper. All of these results have been reported or are expected using other datasets, some by the same authors, e.g. ref 28. A more important issue with this paper is the utility is likely very limited. The authors claim that the small scale design is useful because it can achieve good accuracy without large sample size. I disagree. The PRS framework is useful because one can make prediction without using relatives. When relatives are added, there is no reason to use PRS. In practice, it is not practical to improve the prediction accuracy of one patient by genotyping or phenotyping their relatives unless the relatives are already in a clinical database. Another issue I had with the paper is that the improvement observed isn't a fair comparison. What you should compare is whether adding relatives would improve the prediction of unrelated individuals rather than related individuals in the target set. My suggestion would be to stratify the prediction accuracy in the target set by relatedness and see if any improvement could be observed. These issues seriously undermine my enthusiasm for this paper.

Reviewer #2 (Remarks to the Author):

This study demonstrates that the inclusion of close relatives can lead to large gains in accuracy when predicting complex traits using polygenic scores. It is very interesting to see that predictions from 5000 first-degree relatives can outperform predictions from 220,000 unrelated individuals. The statistical analyses all seem to be sound and conducted very carefully.

My main comments relate to the source of the increase in prediction accuracy.

1. I am uncertain whether it is always the case in polygenic risk prediction that as long as the accuracy increases, it doesn't matter whether the accuracy comes from causal genetic effects or from other sources, such as environmental effects which are picked up by genetic effect estimates. In some application of polygenic risk scores the goal is to only quantify the contribution to risk conferred by causal genetic effects. I think it could be helpful to clarify that the approach of including close relatives will be most useful in applications where accuracy matters more than delineating between causal genetic effects and other effects.

2. Lines 201 to 203 seem to suggest that most of the increase in prediction accuracy can be attributed to the decrease in the effective number of chromosome segments, but this is difficult to tell from the results. A similar, but not identical question is how much of the increase in prediction accuracy can be attributed to familial effects versus causal genetic effects. Figure 3 addresses this question, though it is difficult to translate the fold ratios in prediction accuracy to relative contributions. It's probably outside the scope of this study to dive much deeper into these

questions, but it might be informative for example to include "Unrelated" in Figure 3. This could help by showing how close the empirical prediction accuracies are to theoretical estimates using narrow-sense and family-based h^2 in unrelated samples.

Minor comments:

3. I was confused for a bit by the sentence in lines 206 - 208, which explains that the small difference in M_e between 2nd and 3rd degree relatives was due to the different number of relatives. That seemed to be in conflict with the fact that according to Supp. Table 2, in the small scale design the total number of selected samples should be 6000 for all degrees of relatedness. My understanding now is that when the total number before selection is larger, samples can be selected so that they form fewer, larger families with more relatives per family. If that is the case, it could be useful to report the average number of samples per family in Supp. Table 2.

4. Supp. Figure 2b seems to assume that h^2 is one or close to one, or it measures the accuracy of predicting only the genetic component. It would be good to mention that in the legend.

5. ABLUP is mentioned in the Supplement, but not defined.

6. Line 559 says that the theoretical prediction accuracy depends on effective population size, although it seems that the total number of phenotypic observations is used in practice. It would be good to mention whether the inclusion of close relatives has any effects on the N that should be used in estimating theoretical prediction accuracy.

Best wishes,
Robert

Reviewers' comments:

Reviewer #1 (Remarks to the Author):

This is a useful paper that describes a design in which related individuals are used in the reference panel when making polygenic risk score prediction. The UK Biobank dataset is used to explore the utility of this design. A few findings are reported. First, in a small scale design (6,000 total individuals), the prediction accuracy increases when more related individuals are included between the reference and target sets. However, this improvement is not seen in a large-scale design where the related individuals only constitute a small fraction. Second, the empirical accuracy seems to agree with the theoretical prediction based on family-based heritability, suggesting that familial environmental effect is small. Third, PRS prediction can be achieved the same accuracy with relatives even with small sample size. Fourth, the authors also demonstrated how this could be used in a clinical setting using binary traits. Finally, the prediction can be further improved with ungenotyped relatives.

Response: We thank the reviewer for this nice summary and acknowledgment (a useful paper).

1. Overall, the paper is easy to follow. However, I find nothing new in this paper. All of these results have been reported or are expected using other datasets, some by the same authors, e.g. ref 28.

R: We agree that the theory for the relationship between the prediction accuracy and the effective number of chromosome segments (M_e) has been well established by a number of previous papers including reference #28. We were aware of this and gave the references clearly (ref #25, 26 and 27 on line #64-66 and #365-368). These previous studies were mostly based on simulated data or a small real dataset (e.g. Framingham heart study with ~ 7,000 individuals only) for a limited number of traits (height). However, in this paper, we used biobank scale data with various designs with four different levels of relatedness for 50 complex traits including 12 disease traits. This comprehensiveness allowed us to obtain empirically valid evidences for the usefulness of close relatives. In addition, there are a number of specific observations that have not been reported before, e.g. i) stratified prediction accuracies for 1st, 2nd and 3rd degree relatives (e.g. Figure 2), ii) the comparison between the empirical accuracy and the theoretical accuracy based on family-based heritability especially for biobank scale data (Figures 3), iii) prediction accuracies stratified according to three different categories of 50 traits (Figure 5), iv) clinical impact of polygenic scores for 12 diseases from biobank data (Figure 6), and v) improved prediction accuracy from ungenotyped relatives in human studies (Figure 7).

2. A more important issue with this paper is the utility is likely very limited. The authors claim that the small-scale design is useful because it can achieve good accuracy without large sample

size. I disagree. The PRS framework is useful because one can make prediction without using relatives. When relatives are added, there is no reason to use PRS. In practice, it is not practical to improve the prediction accuracy of one patient by genotyping or phenotyping their relatives unless the relatives are already in a clinical database.

R: We thank the reviewer for this comment to give an opportunity to clarify this issue. It has been shown from the literature that when relatives are added, PRS is still useful. In fact, a number of studies have shown that PRS with relatives outperforms the traditional approach with family information only without their genotypes^{1,2}. We confirm this and show that the use of PRS from GBLUP is better than using only pedigree information (ABLUP) (Supplementary Figure 8), indicating that PRS with relatives can be useful when using bio-bank scale data. It is also noted that PRS with additional relatives is likely to outperform PRS with unrelated individuals only whether using the small- or large-scale design (e.g. Supplementary Figure 2).

In practice, genomic databases such as biobank datasets (e.g. UK, All of us, Estonian, Japanese and Chinese-Kadoori) and commercial genotyping databases (23andMe, Ancestry and MyHeritage) have clinical measures or can be linked with existing national clinical databases with relative information available. Figure R1 shows that the number of genotyped individuals has now increased dramatically³, showing that there is a high probability of finding genotyped relatives of a random sample (e.g. 90% chance of finding at least one third-cousin relative in USA when using only one million samples (please see Khan and Mittelman (2018)³). Therefore, it is likely that the prediction of a random population sample benefits from the information of already known genotypes and phenotypes of their relatives. For example, it will be practical to use all available data and one can select the data that will be most informative, and this paper demonstrates the differences in information of different types of data (based on relatedness). Moreover, future work is required as it is likely that information from other populations or different ethnic groups may actually decrease the accuracy and it is useful to know the difference in value of information.

We have added relevant discussion in line #439-#447.

Figure R1. The estimated number of genotyped customers from 5 companies in 2013-2018³

3. Another issue I had with the paper is that the improvement observed isn't a fair comparison. What you should compare is whether adding relatives would improve the prediction of unrelated individuals rather than related individuals in the target set. My suggestion would be to stratify the prediction accuracy in the target set by relatedness and see if any improvement could be observed. These issues seriously undermine my enthusiasm for this paper.

R: We thank the reviewer for this insightful comment. According to this comment, we used 5000 reference and 2000 target samples in the prediction analysis where the target samples were stratified such that a set of 1000 target samples (T_B) were unrelated to the reference samples (R_1), but the other set of 1000 target samples (T_A) had 1st degree relationships with the reference samples (R_1) (Figure R2a). For a comparison, we also conducted a separate analysis involving two target sets (T_C and T_D) both unrelated to the reference samples (Figure R2b). Note that there were no overlapping individuals between T_A , T_B , T_C and T_D (independent to each other). We obtained prediction accuracies for stratified target samples (e.g. considering T_A or T_B separately) or prediction accuracy for the whole target samples (e.g. $T_A \cup T_B$). Figure R2 shows that using the combined information of unrelated and 1st degree relatives gave ~ 0.2 ($T_A \cup T_B$) while using the information of 1st degree relatives only gave ~ 0.25 (T_A). As expected, using the information of unrelated individuals only gave ~ 0.07 (consistently low accuracies for T_C , T_D , $T_C \cup T_D$, and T_B). It is noted that T_A and T_B were used in the same analysis and same model, and the prediction accuracy for T_A was substantially higher than that for T_B . This result confirms our observations

from the main analyses (Figure 2 in the main text). We added this in the main text line #202-#205.

Figure R2. Prediction accuracies for target datasets stratified by their relatedness to the reference dataset. In the prediction, 5000 reference and 2000 target samples were used, where T_A was a set of 1000 target samples that had 1st degree relationships with the reference samples, and T_B was the other 1000 target samples that were unrelated to the reference samples (a). For a comparison, a separate analysis was considered, in which both target sets (T_C and T_D) were unrelated to the reference samples (b). The error-bars reflect 95% confidence interval (CI).

Reviewer #2 (Remark to the Author):

This study demonstrates that the inclusion of close relatives can lead to large gains in accuracy when predicting complex traits using polygenic scores. It is very interesting to see that predictions from 5000 first-degree relatives can outperform predictions from 220,000 unrelated individuals.

The statistical analyses all seem to be sound and conducted very carefully.

R: We thank the reviewer for emphasizing our key finding.

My main comments relate to the source of the increase in prediction accuracy.

1. I am uncertain whether it is always the case in polygenic risk prediction that as long as the accuracy increases, it doesn't matter whether the accuracy comes from causal genetic effects or from other sources, such as environmental effects which are picked up by genetic effect estimates. In some application of polygenic risk scores the goal is to only quantify the contribution to risk conferred by causal genetic effects. I think it could be helpful to clarify that the approach of including close relatives will be most useful in applications where accuracy matters more than delineating between causal genetic effects and other effects.

R: We thank the reviewer for this insightful comment and agree with the point. We clarified and discussed this point in the 2nd paragraph in Discussion (line #375-#376 in the main text).

“The approach of including close relatives will be most useful in applications where accuracy matters more than delineating between causal genetic effects and other effects.”

2. Lines 201 to 203 seem to suggest that most of the increase in prediction accuracy can be attributed to the decrease in the effective number of chromosome segments, but this is difficult to tell from the results. A similar, but not identical question is how much of the increase in prediction accuracy can be attributed to familial effects versus causal genetic effects. Figure 3 addresses this question, though it is difficult to translate the fold ratios in prediction accuracy to relative contributions. It's probably outside the scope of this study to dive much deeper into these questions, but it might be informative for example to include “Unrelated” in Figure 3. This could help by showing how close the empirical prediction accuracies are to theoretical estimates using narrow-sense and family-based h^2 in unrelated samples.

R: We thank the reviewer for pointing this out. According to the comment, we added Supplementary Figure 3 that shows the actual empirical and theoretical accuracies for analyses with unrelated individuals and 1st – 3rd degree relatives (see Figure R3). We mentioned this in line #228-230 in the main text that the difference between the empirical and theoretical accuracy was proportional to the degree of relatedness.

Figure R3. The actual empirical and theoretical prediction accuracies for analyses with unrelated, 3rd, 2nd and 1st degree relatives when using family-based or narrow-sense heritability. It is noted that family-based heritability for unrelated individuals was estimated based on the small-scale design with unrelated samples. The error-bars reflect 95% confidence interval (CI).

Minor comments:

3. I was confused for a bit by the sentence in lines 206 - 208, which explains that the small difference in M_e between 2nd and 3rd degree relatives was due to the different number of relatives. That seemed to be in conflict with the fact that according to Supp. Table 2, in the small-scale design the total number of selected samples should be 6000 for all degrees of relatedness. My understanding now is that when the total number before selection is larger, samples can be selected so that they form fewer, larger families with more relatives per family. If that is the case, it could be useful to report the average number of samples per family in Supp. Table 2.

R: For the analyses with each level of relatedness, we used a graph and network analysis tool to maximize the average relatedness among selected individuals from the set of each level of kinship relatedness (described in line #508-520). This is why the distribution of relationships depends on the sample size (i.e. the larger the sample size, the higher the number of related pairs). According to the reviewer's comment, we now report the number of individuals per family in Supplementary Table 11 and added in line #521.

4. Supp. Figure 2b seems to assume that h^2 is one or close to one, or it measures the accuracy of predicting only the genetic component. It would be good to mention that in the legend.

R: We suppose your comment was referred to Supplementary Figure 6b (currently Supplementary Figure 2b). We clarified this and added in the legend as “ ..., assuming a heritability of 0.5”.

5. ABLUP is mentioned in the Supplement, but not defined.

R: Thanks for the comment, we updated the manuscript (line #552-554 in the main text).

6. Line 559 says that the theoretical prediction accuracy depends on effective population size, although it seems that the total number of phenotypic observations is used in practice. It would be good to mention whether the inclusion of close relatives has any effects on the N that should be used in estimating theoretical prediction accuracy.

R: We thank the reviewer for spotting this typo. We corrected as “... the sample size in the reference dataset (N)”. The relationship between the effective population size, the number of chromosome segments and pair-wise relationships has been well established (references #25, 26, 27 and 28), and is explicitly described in line #573-#574 in the text.

References

1. Do, C. B., Hinds, D. A., Francke, U. & Eriksson, N. Comparison of Family History and SNPs for Predicting Risk of Complex Disease. *PLoS Genet.* **8**, e1002973.
2. de los Campos, G., Vazquez, A. I., Fernando, R., Klimentidis, Y. C. & Sorensen, D. Prediction of Complex Human Traits Using the Genomic Best Linear Unbiased Predictor. *PLoS Genet.* **9**, e1003608.
3. Khan, R. & Mittelman, D. Consumer genomics will change your life, whether you get tested or not. *Genome Biology* vol. 19 120 (2020).

REVIEWERS' COMMENTS:

Reviewer #1 (Remarks to the Author):

This is a revision of a previously reviewed manuscript. My biggest concern was as follows:

"Another issue I had with the paper is that the improvement observed isn't a fair comparison. What you should compare is whether adding relatives would improve the prediction of unrelated individuals rather than related individuals in the target set. My suggestion would be to stratify the prediction accuracy in the target set by relatedness and see if any improvement could be observed."

The authors now present data to stratify prediction accuracy in the target set. The fair comparison in the new figure (Figure S1 and in response to reviewer) is comparing T_B with T_D or T_C. The data now confirmed my suspicion, i.e. adding related individuals only increase prediction for related individuals but not unrelated individuals. This greatly limit the utility of this approach and something that has been known for a very long time. All the improvement you see isn't a result of improved study design but an artifact that related individuals were used to evaluate accuracy. Thus, I don't think this paper is acceptable.

Reviewer #2 (Remarks to the Author):

The authors have addressed and resolved all points which I have raised in the initial review, and I have no more reservations.

Response to Referees

We thank the reviewers for comprehensive and insightful comments in the review process that help for us to improve the manuscript. We give point-by-point response to the remaining concern below.

REVIEWERS' COMMENTS:

Reviewer #1 (Remarks to the Author):

This is a revision of a previously reviewed manuscript. My biggest concern was as follows:

"Another issue I had with the paper is that the improvement observed isn't a fair comparison. What you should compare is whether adding relatives would improve the prediction of unrelated individuals rather than related individuals in the target set. My suggestion would be to stratify the prediction accuracy in the target set by relatedness and see if any improvement could be observed."

The authors now present data to stratify prediction accuracy in the target set. The fair comparison in the new figure (Figure S1 and in response to reviewer) is comparing T_B with T_D or T_C . The data now confirmed my suspicion, i.e. adding related individuals only increase prediction for related individuals but not unrelated individuals. This greatly limit the utility of this approach and something that has been known for a very long time. All the improvement you see isn't a result of improved study design but an artifact that related individuals were used to evaluate accuracy. Thus, I don't think this paper is acceptable.

Response: We do not agree with that the improved accuracy was due to artifact. As the accuracy is a function of the effective number of chromosome segments, heritability and sample size (eq. 1; Figures 2, 3 and 4), the low prediction accuracy for T_B in contrast to T_A (Supplementary Figure 1) is well expected from theory, i.e. the increase in accuracy is limited to the samples in the target set that do have close relatives in the discovery set. We made this clearer in line #216. "The lower prediction accuracy for T_B was expected because of the fact that the increase in accuracy is limited to the samples in the target set that do have close relatives in the discovery set."

Reviewer #2 (Remarks to the Author):

The authors have addressed and resolved all points which I have raised in the initial review, and I have no more reservations.

Response: We thank the reviewer for valuable and constructive comments in the previous version.